# Lossless Image Steganography Based on Invertible Neural Networks

**DOI:** 10.3390/e24121762

**Published:** 2022-12-01

**Authors:** Lianshan Liu, Li Tang, Weimin Zheng

**Affiliations:** College of Computer Science and Engineering, Shandong University of Science and Technology, Qingdao 266590, China

**Keywords:** steganography, invertible neural networks, lossless recovery, deep learning

## Abstract

Image steganography is a scheme that hides secret information in a cover image without being perceived. Most of the existing steganography methods are more concerned about the visual similarity between the stego image and the cover image, and they ignore the recovery accuracy of secret information. In this paper, the steganography method based on invertible neural networks is proposed, which can generate stego images with high invisibility and security and can achieve lossless recovery for secret information. In addition, this paper introduces a mapping module that can compress information actually embedded to improve the quality of the stego image and its antidetection ability. In order to restore message and prevent loss, the secret information is converted into a binary sequence and then embedded in the cover image through the forward operation of the invertible neural networks. This information will then be recovered from the stego image through the inverse operation of the invertible neural networks. Experimental results show that the proposed method in this paper has achieved competitive results in the visual quality and safety of stego images and achieved 100% accuracy in information extraction.

## 1. Introduction

In recent years, steganography has become the focus of information security technology. With the popularization of the Internet and the development of multimedia technology, media such as books, audio, video, and images have been continuously copied and rapidly disseminated, enriching people’s lives. However, this is followed by increasingly rampant information eavesdropping, infringements, and piracy activities, which seriously damage the legitimate rights and interests of creators and cause people to worry about information security issues. In order to solve these security problems in the network environment, scientists first use encryption technology to encrypt the information to be transmitted and then send the encrypted information out. The receiver receives the encrypted data and decrypts the data through the decryption algorithm designed by the sender, thereby restoring the original communication data. This method makes it impossible for eavesdroppers to understand the eavesdropped information, thereby improving the security of network information transmission. However, encryption technology turns the original information into meaningless messy information, which attracts the attention of attackers. In order to solve this problem, information-hiding technology has been proposed and has attracted widespread attention from researchers.

Steganography is a scheme that makes it impossible for anyone other than the intended recipient to know the event of the information transmission or the content of the message by embedding secret information into the cover image. Unlike cryptography, which makes information unreadable through different transformations of the text, steganography also hides the existence of information [1]. Image steganography embeds secret information into the cover image without destroying the original image, making the stego image visually consistent with the cover image, thus avoiding attacks. Today, steganography has been used in many areas of the digital world, such as copyright protection, information appending, tamper-proofing, and classified communications [2,3]. However, in special fields such as military, medicine, and law, people put forward higher requirements for the accuracy of information because of the consideration of confidentiality, security, and fairness. That is to say, special fields require lossless steganography to completely extract hidden secret information.

A good steganography approach pays more attention to the visual quality, hidden capacity, recovery accuracy, and security of the stego image, which is slightly different from a digital watermark, which focuses more on robustness and nonremovability. Traditional steganography methods typically use the least significant bit method to hide secret information in the spatial domain [4,5] and the frequency domain [6,7,8] of the image or adaptively select the embedding region [9] by analyzing the texture complexity of the image. Among them, many methods can achieve the lossless recovery of message [10,11], but the hidden capacity is small and the antianalysis ability is insufficient.

With the popularity of deep neural networks (DNNs) [12], steganography has also developed rapidly. Early artificial neural networks (ANNs) [13,14] mimic biological neural networks, treating neurons as a logical unit, and this unit usually consists of three parts: input layer, hidden layer, and input layer. And its main role is to map manually selected features to values to achieve data classification. DNNs do not need to manually extract features but instead automatically combine the underlying features of the data to form more-abstract high-dimensional features, which greatly promotes the development of computer vision and natural language processing. A convolutional neural network (CNN) [15,16] is one of the representative algorithms of DNNs, which adds a feature-learning part on the basis of the original artificial neural network and uses the convolutional layer to automatically extract the local features of the image. Because CNNs are applied in the field of image steganography, researchers no longer need to manually identify and select the appropriate embedding region, and the neural network will automatically select the best position for data embedding through learning, thereby improving the embedding capacity and antidetection ability of stego images. Baluja [17,18] was the first to apply CNN to image steganography, and the proposed model trained three convolutional neural networks, which were used to preprocess secret image, hide image in cover image, and reveal hidden image, separately. However, if the model uses different networks for encoding and decoding, separately, the relationship between them tends to be loose, which can easily lead to color distortion and artifacts in stego images. To solve this problem, generative adversarial networks (GANs) [19] were introduced into stego-image-generation frameworks. GAN is an artificial intelligence algorithm designed to solve image-generation problems, and unlike most other generative models based on optimization ideas, GANs are based mainly on game theory. In the field of information hiding, researchers have taken advantage of the competitiveness between generators and discriminators to generate stego images with better visual quality [20,21,22,23]. In addition, researchers have used residual networks [24] to improve the network structure of generators and discriminators [25], which improved the security and robustness of algorithms. No-coverage steganography [26,27,28,29] and no-embedding steganography [30,31] have also been proposed, further enhancing the robustness and antianalysis of stego images. Most of the above methods tend to improve the visual effect and security of stego images while ignoring the recovery accuracy of secret information.

To solve the problem that the current steganography cannot be extracted without loss, a steganography method based on invertible neural networks is proposed to extract embedded secret information without considerable loss. This paper uses invertible neural networks for the embedding of secret information, which ensures good visual effects and provides an antidetection ability for stego images. The extraction process uses the same network structure and network parameters as when encoding, so most models based on invertible neural networks are lightweight. In addition, because invertible neural networks prevent information loss, they retain details of the input data. Ideally, the extraction of secret information and the recovery of cover images will be completely lossless without taking into account the rounding error when saving stego images. However, the rounding error of saving the image cannot be avoided, and there will be some differences between the original information and the secret information extracted by the reverse calculation when using the invertible neural networks. This means that the lossless extraction cannot be realized. Therefore, we propose converting the secret information into binary sequence and then embed it into cover image and use the appropriate threshold to recover the secret information while extracting and to prevent significant loss. In addition, we design a mapping module to increase the hidden capacity while ensuring 100% recovery for the message.

## 2. Related Work

### 2.1. Steganography

According to the working domain and the application scope of the steganography algorithm, the traditional image steganography can be divided into three categories: steganography algorithm based on spatial domain, steganography algorithm based on transformation domain, and adaptive steganography algorithm. The most commonly used algorithm in the spatial domain is the least significant bit algorithm [32,33], which embeds the binary sequence into the lowest k significant bits of cover image. However, this method can easily produce artifacts in stego images and can be judged as stego images by steganalysis algorithm [34]. In order to solve the problem of poor robustness and low security for spatial domain steganography, researchers consider embedding information in the transformation domain. Discrete cosine transforms [8,35,36] and wavelet transforms [6,32,37,38] are typically used to convert images from the spatial domain to the transformation domain and then convert the image back into the spatial domain after embedding the message. The information-hiding method of transform domain is improved in terms of robustness and antidetection ability compared with the spatial domain. The adaptive steganography algorithm [39,40,41,42] selectively embeds secret information into areas with more-complex textures and more-obvious edges by analyzing the texture characteristics of the cover image, which ensures a good visual effect for the stego image and improves the ability of resisting detection.

Recently, with the vigorous development of deep learning, some researchers have applied deep-learning networks to image steganography. Among them, Rehman [43] proposed to train the end-to-end network for image steganography and designed an encoder and a decoder to generate stego images and to extract secret information. Compared with the traditional steganography method, Rehman [43] implemented embedding gray-scale images into color images of the same size. As a result, the hiding capacity was greatly improved, but the resulting stego images experienced obvious color distortion. Hayes [44] then proposed adding a discriminator to the end-to-end network to distinguish whether an image was a stego image or a cover image and to improve the performance of the generator and discriminator through adversarial training. On the basis of previous research, Tang [45] proposed a novel operation, which could deceive analyzers with convolutional neural network structures while embedding secret information, achieving better security performance. Yang [46] applied the U-Net network architecture in the generator and combined it with the generation adversarial networks, achieving a more advanced effect. Liao [47] utilized the correlation of RGB channels to adaptively allocate the embedded capacity of each channel, achieving better performance in resisting modern color image steganalysis. Hu [30] pointed out that embedded-based image steganography inevitably left traces of modification that would be detected by increasingly advanced steganalysis algorithms. Therefore, Hu [30] proposed a nonembedded steganography that did not need to modify the cover image to improve the antianalysis ability of stego images. In addition, in order to obtain greater hidden capacity, Baluja [17] first proposed embedding a color image entirely into another color image of the same size by using deep neural networks and then proposed to improve the network structure to enhance the security performance of stego images [18]. The above neural network–based methods usually have a good antidetection ability, but few methods can achieve lossless extraction for secret information.

### 2.2. Invertible Neural Networks

In recent years, invertible neural networks had attracted the attention of many researchers because of their excellent performance. The invertible neural network maps a high-dimensional complex distribution Px to a simple latent distribution Pz through a series of reversible transformations and utilizes the neural network to learn the mapping relationship between distributions Px and Pz. The forward process of the framework takes the high-dimensional complex data x as the input and outputs data z that conforms to a simple distribution. The reverse process is the generation process of a generative model, taking the sampled data z as input to generate high-dimensional complex data x. Dinh [48] was the first to propose invertible neural networks, and the basic architecture of which was the affine coupling layer popularized by the RealNVP [49] model. In addition, Dinh [49] introduced convolutional layers in the coupled model to suit image processing tasks. On the basis of invertible neural networks, Kingma [50] proposed Glow, a simple type of generative stream using an invertible 1 × 1 convolution and was capable of efficiently synthesizing large and subjectively realistic images. The basic working principle of invertible neural networks is to divide the input data into two parts, which are converted by arbitrarily function and are coupled in an alternating manner. However, the arbitrary function does not need to be reversible by itself. A deeply invertible network consists of a series of the above building blocks. Invertible neural networks have three characteristics: the map from input to output is bi-beamed, the size of the network of input and output must be consistent, and the Jacobian of the network is not equal to 0.

There had been several studies that applied invertible neural networks to image steganography. Xu [51] proposed a learning-based approach that effectively improved the robustness of stego images while maintaining their imperceptibility and large capacity. Jing [52] applied invertible neural networks combined with a wavelet transform to the hiding of a single color image, which had greatly improved the visual effect and extraction accuracy compared with the traditional method of embedding and extraction using two networks, separately. Lu [53] proposed hiding multiple color images in a color image of the same size, which greatly increased the hidden capacity of image steganography. Guan [54] added an importance map module on the basis of invertible neural networks, which guided the hiding of the next image with the hidden results of the current image, avoiding the problems of contour shading and color distortion caused by multi-image hiding. This also improved the invisibility of secret information. The above methods are all more relevant to the hidden capacity of the image rather than lossless extraction of secret information. In addition, invertible neural networks have a wide range of applications in the areas of image compression [55], image rescaling [56,57], image-to-image translation [58], and video superresolution [59].

## 3. Methods

### 3.1. Overview

The purpose of the steganography method proposed in this paper is to generate high-quality stego images and extract embedded secret information without loss. As is shown in Figure 1, the method of this paper converts the secret information into binary sequence and then uses invertible neural networks to hide the binary sequence in the cover image, generating the stego image and the key file z. The inverse transformation takes the rounded image stego’ and key file z as input and extracts secret_rev and cover_rev through the reverse operation of the invertible neural networks. Because of the inevitable rounding error when the stego is saved, the secret information cannot be accurately restored after the reverse operation. This means that the extracted secret_rev is not a binary sequence containing 0 and 1. To solve this problem, we set a threshold t that maps the value greater than t in the secret_rev to 1 and the value less than t to 0, thus obtaining the secret information consistent with the message. In addition, we add a mapping module map, which increases the hidden capacity of the method from 1 bpp to 3 bpp by compressing the secret information.

### 3.2. Network Architecture

Figure 1 shows the overall network architecture proposed in this paper. Unlike many of the methods that currently exist, the embedding and extraction process proposed in this paper uses the same network, which is INNs (invertible neural networks) proposed by Dinh [48]. On the one hand, this can avoid the laborious process of having to train the two networks of encoder and decoder, thus reducing the workload of training the model. On the other hand, the INN mentioned in this paper essentially uses the neural network to scale the value of the secret information and then adds the corresponding position to the cover image. This ensures the natural similarity between the stego image and the cover image.

The *η*, *ρ*, and *φ* in reversible blocks can be arbitrarily complex functions. In this paper, we use three widely used dense blocks [60] to represent them. Different from the residual network that establishes a skip connection between the front layer and the back layer, the dense blocks establish a dense connection between all the previous layers and the back layer, and the feature reuse is achieved through the connection of features on channel. The specific structure is shown in Figure 2. The mapping module is designed to improve extraction accuracy. If the sum of all the values of the binary information is greater than 1/2 of its length, the mapping module will interchange the values of 0 and 1, thereby reducing the amount of information actually embedded and increasing the hidden capacity of the method proposed by this paper. Correspondingly, when the secret information is extracted, the threshold t remains unchanged, the value greater than t in the secret_rev is mapped to 0, and the value less than t is mapped to 1. In the experiment, we find that when the capacity increases to 3 bpp, then even if the above mapping method is used, it is difficult to achieve a loss-free extraction; that is, the maximum error of extracting information will exceed the threshold t, which will lead to errors in the extraction process. Therefore, we try to map the binary data 0 and 1 to 0 and 0.5, which further reduces the amount of actual embedded information while setting the threshold to 0.25. The experiment proves that this method is feasible. The mapping process is detailed in Algorithms 1 and 2.
**Algorithm 1:** Mapping module map**Input:** message**Output:** secret_map**Initialization**t = 0.25; turn = false;**If**(sum(message) > len(message)/2) **then** secret_map = message < t turn = True**else** secret_map = message**end if**secret_map = secret_map/2
**Algorithm 2:** Inverse mapping module i_map**Input:** secret_rev**Output:** message_rev**If**(turn) **then** message_rev = secret_rev < t**else** message_rev = secret_rev >= t**end if**

As is shown in Figure 1, our network uses N reversible blocks of the same structure, which is the affine coupling layer proposed by Dinh [49]. Among them, the first reversible block takes the cover image and secret_map as input, outputting cover1 and secret1. Each subsequent reversible block takes the output of the previous reversible block as input, and the last reversible block outputs the final result stego and key z. The specific calculation process is as follows:(1)$${\mathrm{cover}\;}_{i+1}={\mathrm{cover}\;}_{i}+\varphi {(\sec\mathrm{ret}\;}_{i})$$
(2)$${\sec\mathrm{ret}\;}_{i+1}={\sec\mathrm{ret}\;}_{i}\;⊙\;\exp(\rho {(\mathrm{cover}\;}_{i+1}))+\eta {(\mathrm{cover}\;}_{i+1})$$
where φ(•), ρ(•), and η(•) represent the three dense blocks and  ⊙ represents the Hadamard product. In the reverse operation of a reversible block, the input is the output of the forward operation, which are the stego and the key z. It can be derived from its reverse calculation process from Equations (1) and (2):(3)$${\sec\mathrm{ret}\;}_{i}={(\sec\mathrm{ret}\;}_{i+1}-\eta {(\mathrm{cover}\;}_{i+1}))⊙\exp(-\rho {(\mathrm{cover}\;}_{i+1}))$$
(4)$${\mathrm{cover}\;}_{i}={\mathrm{cover}\;}_{i+1}-\varphi {(\sec\mathrm{ret}\;}_{i})$$
After the last inverse transformation, the extracted secret information secret_rev is mapped to a sequence of 0 and 1 according to the threshold.

### 3.3. Loss Function

The overall loss function of this paper consists of two parts: the loss caused by the difference between the stego image and the cover image, and the loss caused by the difference between the secret_rev and the secret_map.

In forward computation, we expect the stego image to be as similar as possible to the cover image. This guarantees good visuals and avoids detection by steganography, thereby improving its safety. The calculation process for hidden losses is as follows:(5)$$Loss_{\mathrm{enc}}=\sum_{n=1}^{N}L({\mathrm{cover}}^{\left(n\right)},{\mathrm{stego}}^{\left(n\right)})$$
where *N* refers to the number of training images in a batch, and *L* is used to measure the difference between cover image and stego image. We use the l2 distance here. The l2 distance, also known as the Euclidean distance, is the square root of the sum of squares of all elements of a vector.

In the reverse calculation process, we expect the secret_rev to be as similar as possible to the secret_map, ensuring that the error between the two is within the appropriate range. In this way, the message will be restored without loss. The calculation process for the extraction loss is as follows:(6)$$Loss_{\mathrm{dec}}=\sum_{n=1}^{N}L(\sec\mathrm{ret}\_{\mathrm{map}}^{\left(n\right)},\sec\mathrm{ret}\_{\mathrm{rev}}^{\left(n\right)})$$
The overall loss is calculated as follows:(7)$$Loss=\alpha Loss_{enc}+\beta Loss_{dec}$$
where *α* and *β* are the parameters that balance the two losses.

## 4. Experiments

### 4.1. Experimental Settings

The proposed method uses two datasets, coco2017 [61] and pascalvoc2012 [62]. There are five annotation types in the coco dataset, and the pascalvoc dataset consists of 20 categories. We randomly select 1000 images as the training set in the coco2017 dataset, and use 40,670 images from the coco2017 test set and 17,124 images from pascalvoc2012 to evaluate the method. During the training and the testing process, we use the center cropping strategy to crop the image to a size of 128 × 128. In this experiment, we set the number of reversible blocks to 8, the batch size to 2, and the parameters of the population loss function *α* and *β* to 1 and 10, respectively.

We use three steganography methods as a contrast, one of which is the traditional steganography method, LSB-ibit; the next is the most commonly used steganography algorithm, Rehman [43], which contains encoding and decoding networks; and the third one is steganography algorithm HiNet, which is based on invertible neural networks [52]. In order to ensure the visual quality of the stego image, we set the embedding capacity of the LSB method to 1 bpp. In addition, Rehman [43] and HiNet [52] were originally used to hide images, where Rehman [43] was used to hide one gray-scale image and HiNet [52] was used to hide one color image. In order to ensure that the above two methods are consistent with the message-hiding configuration in this paper, we make some modifications to the input structure of these two network models so that the data they embed are binary messages. Finally, for a relatively unbiased comparison, we train and test Rehman [43] and HiNet [52] using the same datasets mentioned in this paper.

In this paper, the effect of the method is evaluated from four aspects: visual quality, embedding capacity, recovery accuracy, and detection accuracy. Among them, the visual quality is evaluated by the PSNR (peak signal-to-noise ratio) and the SSIM (structural similarity) index. Embedding capacity bpp refers to the number of bits embedded per pixel. RA (recovery accuracy) refers to the degree of similarity of message_rev to message, that is, the ratio of the equal number of bits of the two binary sequences to their length. Because the data extracted by HiNet [52] and Rehman [43] are not the integers 0 and 1, we round the extracted result. DA (detection accuracy) refers to the correct rate at which an image is determined to be a stego image using the steganalysis algorithm.

### 4.2. Comparison

Table 1 shows the PSNR and SSIM of LSB-1bit, Rehman [43], HiNet [52], and our method. As is shown in the table, both indicator values of our method are significantly better than the others. Specifically, on the coco2017 and pascalvoc2012 datasets, the PSNR of our method improves by 11.4114 db and 13.3013 db, respectively, compared with the second-best-performing results. This is because the proposed method compresses the input data, reduces the actual number of data embedded in the cover image, and greatly improves the hiding performance of the model. So we can achieve better results than the other three methods can. In addition, we can see similar improvements in the SIMM.

Table 2 demonstrates the embedding capacity (C) and recovery accuracy (RA) of LSB-1bit, Rehman [43], HiNet [52], and our method on the coco2017 and pascalvoc2012 datasets. As is shown in the table, the recovery accuracy of Rehman [43] is less than 60, while the HiNet [52] method has been greatly improved over Rehman [43]; LSB-1bit can achieve lossless recovery for secret information on both the coco2017 and the pascalvoc2012 datasets; and our method improves the hidden capacity and antidetection ability of images while ensuring lossless recovery. Among them, Rehman [43] and HiNet [52] cannot fully extract the information, because the secret information undergoes complex linear and nonlinear calculations during embedding and extraction, resulting in inevitable rounding errors. Further, our method can achieve lossless recovery of information because of the reversibility of the invertible neural network itself and, on the other hand, because the mapping module can restore the extracted data to the original binary data through the suitable threshold t. The combination of the above two modules enables our method to achieve a complete extraction of information.

Figure 3 shows the visual effect of stego-image-embedded secret information using our method and the residual plot between the stego image and the cover image. The residual plots of the fourth column are the results of 40× magnification, and the residual plots of the fifth column are the results of 200× magnification. As is shown in the figure, the stego image and the cover image of the proposed method are difficult to distinguish with naked human eye, and the residual plot between the two is almost purely black, which shows that the difference between our stego image and the cover image is extremely small.

Figure 4 shows the stego images generated by LSB-1bit, Rehman [43], HiNet [52], and our method, as well as the residual plot between the stego image and the cover image. The residual plots of the fourth row are the results of 40× magnification. As is shown in the figure, the HiNet [52] and the LSB-1bit have no obvious visual differences from the cover image. The stego image of the Rehman [43] have more-obvious color distortion, the residual plot of which is also the most pronounced. Compared with the above three methods, our method not only recovers secret information without loss but also shows outstanding performance in visual effects. This is also mainly due to the compression of the input data, which reduces the actual number of embedded data, thus minimizing the residual between the stego image and the cover image.

### 4.3. Steganalysis

Steganalysis is the method of measuring the security of stego images. The mainstream steganalysis methods can be divided into statistical steganalysis methods and deep-learning-based steganalysis methods. We use the open-source tool StegExpose for statistical steganalysis. Specifically, we constantly modify the detection threshold on a scale of 0 to 1 to get different true positive rate and false positive rate values and connect all the resulting points into an ROC curve. In Figure 5, we can see that StegExpose does not work well on our method and that the detection accuracy is close to random guessing. This shows that the stego images generated by our method are highly secure and can resist the detection of traditional steganalysis.

SRNet [63] is a steganography detector based on deep convolutional neural networks, to distinguish the stego image from the cover image. The closer the test result is to 50%, the better the steganography performs. The results of the steganalysis are shown in Table 3, from which we can see that the steganalysis results of the LSB-1bit method, the Rehman method [43], and the HiNet method [52] on both datasets are all above 96%, which suggests that the SRNet method can easily distinguish whether the image is a cover image or a stego image. However, our method has a detection rate of less than 52% on both datasets, which illustrates that our method is more secure than the other three methods.

### 4.4. Ablation Experiment

The mapping module proposed plays a key role in recovery accuracy and embedding capacity. In Table 4, w/ mapping means that the model contains a mapping module, and w/o mapping means that there is no mapping module in the model. As is shown in this table, when the embedding capacity is 1 bpp, it can be extracted without loss because the amount of embedded information is small enough. When we increase the embedding capacity to 3 bpp, only the addition of a mapping module can achieve 100% recovery accuracy, which illustrates the effectiveness of the proposed method.

## 5. Conclusions

In this paper, we proposed an image steganography framework on the basis of invertible neural networks, which used forward and backward propagation of the same network to complete the embedding and extraction of information and achieve good visual effects for stego images and lossless recovery for secret information. Among them, we designed a mapping module to convert messages or any other type of data into binary form, and then we compressed the binary sequence to increase the hidden capacity of the method. Comprehensive experiments demonstrated that our method is highly competitive in extraction accuracy, safety, and visual effects, and it significantly both quantitatively and qualitatively outperforms other existing methods.

## Figures and Tables

**Figure 1 entropy-24-01762-f001:**
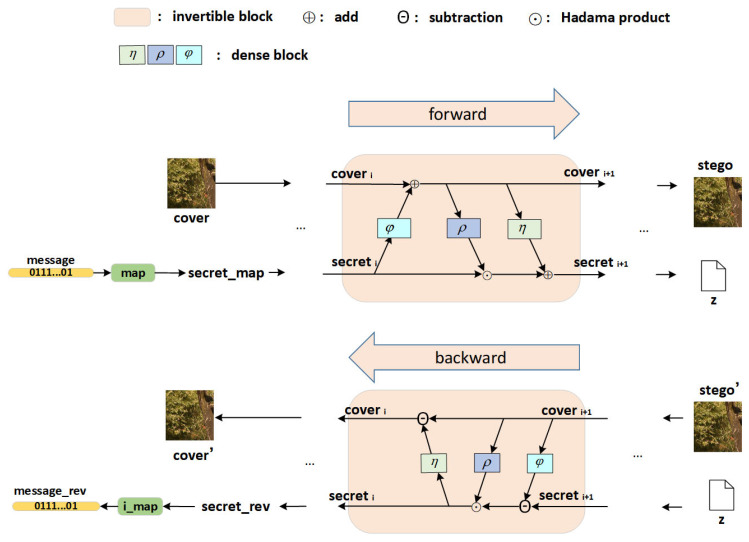
Generation and extraction process. The *η*, *ρ*, and *φ* are three dense blocks, and the exp(·) of *ρ* is omit.

**Figure 2 entropy-24-01762-f002:**
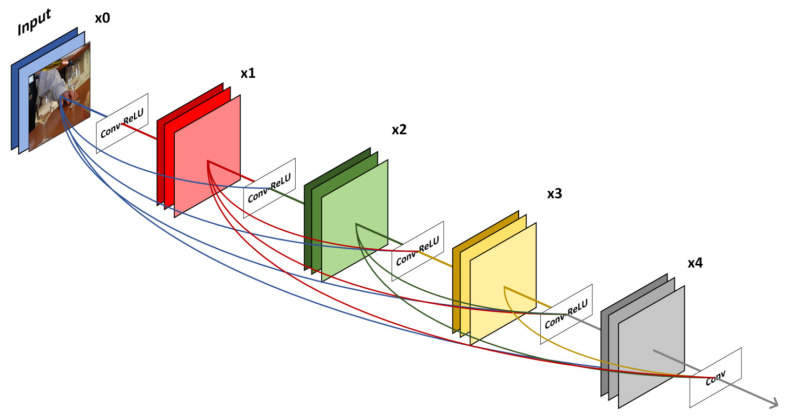
The structure of dense blocks *η*, *ρ*, and *φ* used in our framework. Each layer takes all preceding feature maps as inputs.

**Figure 3 entropy-24-01762-f003:**
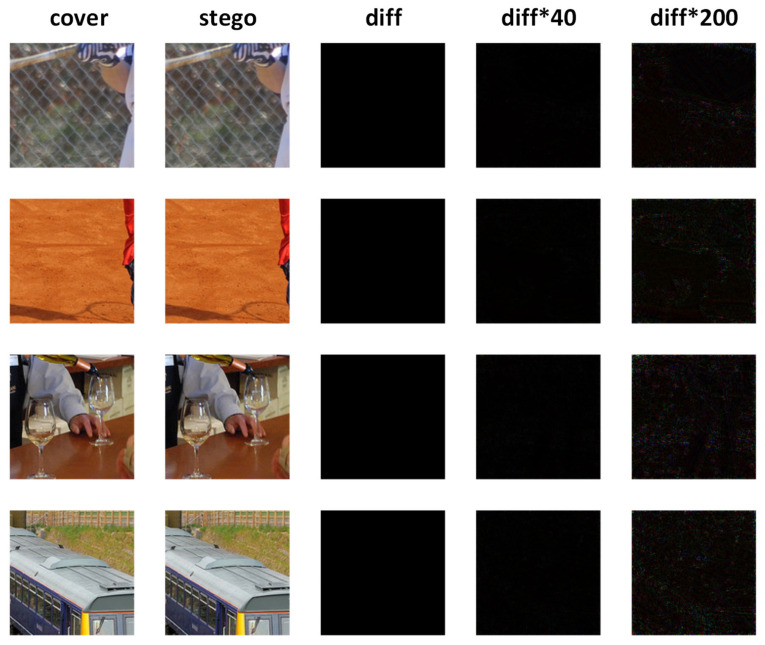
Residual plots of the cover image and the stego image using our method.

**Figure 4 entropy-24-01762-f004:**
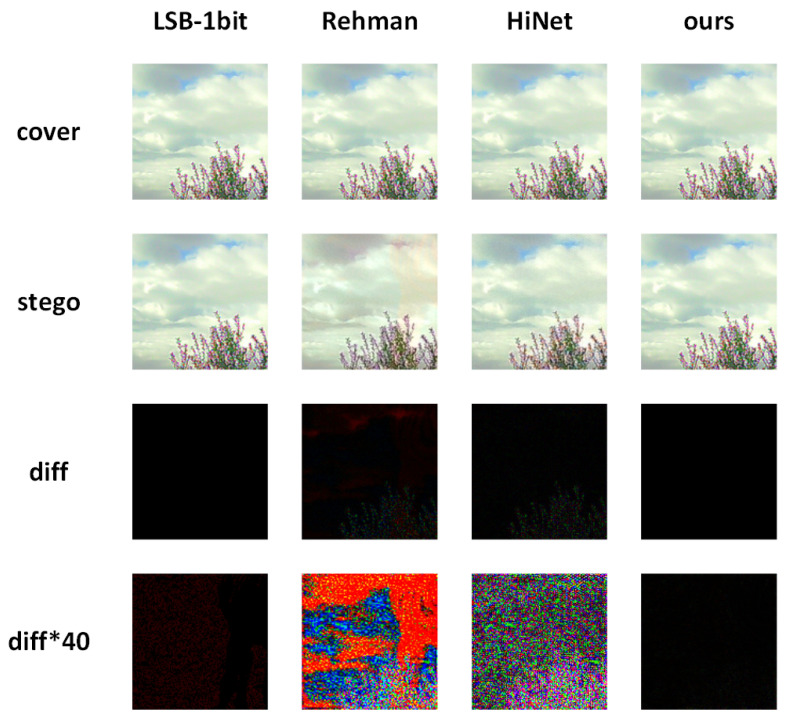
Comparison of residual plots of the cover images and the stego images using LSB-1bit, Rehman [43], HiNet [52], and our method.

**Figure 5 entropy-24-01762-f005:**
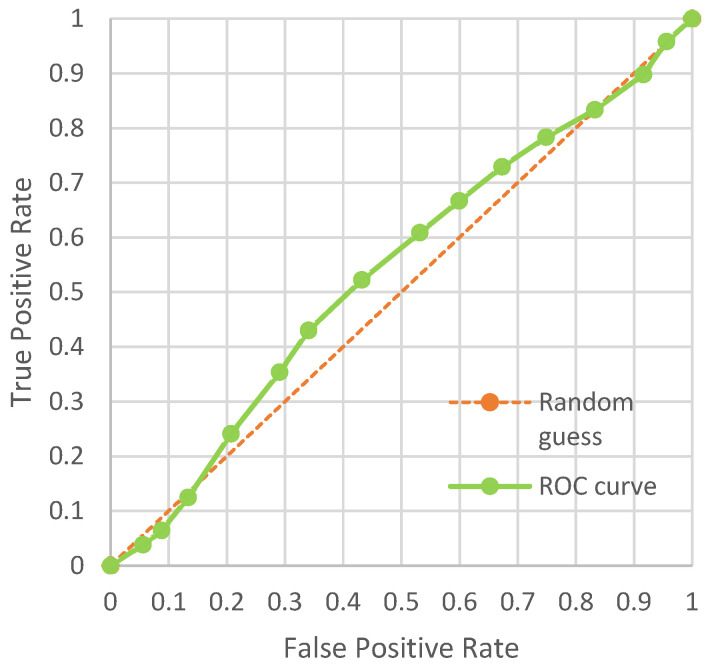
The ROC curve produced by StegExpose for our method.

**Table 1 entropy-24-01762-t001:** **The** PSNR and SSIM for different approaches to the coco2017 and pascalvoc2012 datasets.

Methods	Cover/Stego			
	coco2017		pascalvoc2012	
	PSNR	SSIM	PSNR	SSIM
LSB-1bit	55.8896	0.9993	55.9129	0.9994
Rehman [43]	41.4314	0.9982	38.7726	0.9976
HiNet [52]	43.7672	0.9923	42.7289	0.9948
Ours	67.3010	0.9999	69.2142	0.9999

**Table 2 entropy-24-01762-t002:** Hidden capacity and recovery accuracy for different methods.

Methods	Secret/Secret_rev		
	C (bpp)	RA (%)	
		coco2017	pascalvoc2012
LSB-1bit	1	100	100
Rehman [43]	1	99.83	99.84
HiNet [52]	3	78.59	78.33
Ours	3	100	100

**Table 3 entropy-24-01762-t003:** The detection accuracy of different methods using SRNet.

Methods	DA (%)	
	Coco2017	pascalvoc2012
LSB-1bit	97.15	96.79
Rehman [43]	98.67	99.75
HiNet [52]	99.84	99.77
Ours	51.62	51.81

**Table 4 entropy-24-01762-t004:** The validity of the map module.

Methods	w/ Mapping	w/o Mapping
	RA (%)	RA (%)
Ours (1 bpp)	100	100
Ours (3 bpp)	100	99.48

## Data Availability

Not applicable.

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
