# Peer review of "Lossless Image Steganography Based on Invertible Neural Networks"

_entropy, 2022, doi:10.3390/e24121762_

Round 1
Reviewer 1 Report
The paper proposed the lossless Image Steganography based on ANN. This paper is not suitable for publication due to the following reasons:
(a) There is no introduction for ANN considering this paper uses ANN.
(b)There is no contribution of the paper.
(c) This paper does not provide the fundamental of the ANN. There is no pseudocode and etc.
(d) The governing equation is weak. it seems like the ANN part is not shown thoroughly.
(e) There is no schematic diagram of the neuron.
(f) The equation (1)-(4) is badly written
(g) There is no source of dataset.
(h) The author should use statistical test to validate the superiority of the proposed method/
(i) The analysis is very shallow. The discussion is only talk about trend.
Reviewer 2 Report
Report atacahed

Author Response
Response to Reviewer 2 Comments
Point1: I therefore find that this article may be published without change
Response1: Thank you very much for the affirmation of our manuscript.
Reviewer 3 Report
The proposal presented in this article, in addition to being interesting, is well-presented, sustained, and justified.
The authors proposed a solution for a current steganography problem, once it can not be extracted without loss.
The solution is a steganography method based on invertible neural networks with the aim to extract embedded secret information without considerable loss.
Based on the presented results, compared to the existing steganography method, the recovery accuracy and security had achieved competitive results, and the same happened with the visual effect of stego images had also been greatly improved.
I would like to take this opportunity to congratulate the authors and suggest conclusions improvement in order to be supported by the results presented; and reading the article for minor corrections.
Round 2
Reviewer 1 Report
The author has made some modifications to the paper. Unfortunately, this paper is still far away from the optimal quality of a high-impact journal. This is due to the following reasons:
1. The introduction is very shallow. There is no objective of the paper. In other word, there is no scope or aim why the paper is presented this way.
2. All the figures proposed by the author are very low standards and very badly constructed.
3. The governing equation of the proposed ANN is out of order. It shows that the author is not serious with the correction
4. Again the discussion is badly written. This level of discussion should not be accepted.
5. The pseudocode is badly written. It was all over the place.
Round 3
Reviewer 1 Report
The paper has improved. My concern about the paper is that this paper does not highlight the discussion on the earlier Artificial Neural Network.
